# Genome-Wide Identification and Analysis of *MYB* Transcription Factors in *Pyropia yezoensis*

**DOI:** 10.3390/plants12203613

**Published:** 2023-10-19

**Authors:** Xinzi Yu, Lei Tang, Xianghai Tang, Yunxiang Mao

**Affiliations:** 1MOE Key Laboratory of Marine Genetics and Breeding, College of Marine Life Sciences, Ocean University of China, Qingdao 266003, China; 2MOE Key Laboratory of Utilization and Conservation of Tropical Marine Bioresource & Yazhou Bay Innovation Institute, Hainan Tropical Ocean University, Sanya 572022, China; 3Laboratory for Marine Biology and Biotechnology, Qingdao National Laboratory for Marine Science and Technology, Qingdao 266237, China

**Keywords:** transcription factors, *MYB* superfamily, red algae, *Pyropia yezoensis*, phylogenetic analysis, gene function

## Abstract

*MYB* transcription factors are one of the largest transcription factor families in plants, and they regulate numerous biological processes. Red algae are an important taxonomic group and have important roles in economics and research. However, no comprehensive analysis of the *MYB* gene family in any red algae, including *Pyropia yezoensis*, has been conducted. To identify the *MYB* gene members of *Py. yezoensis*, and to investigate their family structural features and expression profile characteristics, a study was conducted. In this study, 3 *R2R3*-*MYB*s and 13 *MYB*-related members were identified in *Py. yezoensis*. Phylogenetic analysis indicated that most red algae *MYB* genes could be clustered with green plants or Glaucophyta *MYB* genes, inferring their ancient origins. Synteny analysis indicated that 13 and 5 *PyMYB* genes were orthologous to *Pyropia haitanensis* and *Porphyra umbilicalis*, respectively. Most Bangiaceae *MYB* genes contain several Gly-rich motifs, which may be the result of an adaptation to carbon limitations and maintenance of important regulatory functions. An expression profile analysis showed that *PyMYB* genes exhibited diverse expression profiles. However, the expression patterns of different members appeared to be diverse, and *PyMYB5* was upregulated in response to dehydration, low temperature, and *Pythium porphyrae infection*. This is the first comprehensive study of the *MYB* gene family in *Py. Yezoensis* and it provides vital insights into the functional divergence of *MYB* genes.

## 1. Introduction

As one of the three ancient plant groups, red algae originated from a primary photosynthetic endosymbiosis event and donate plastids that form numerous other photosynthetic lineages. Thus, red algae have contributed dramatically to broader eukaryotic evolution and diversity [1]. Some red algae also have important economic value and create important economic results. *Pyropia yezoensis* (nori) belongs to Bangiales (Rhodophyta), which is the most genetically diverse order of red algae and is a principal species used in the economic cultivation of edible seaweed [2].

Transcription factors (TFs), such as *MYB*, *WRKY*, and *bZIP*, play essential roles in plant molecular stress regulation [3]. Among the various TF families in the genome, the *MYB* family is one of the largest and most functionally diverse families [4]. Thus, it is usually called the *MYB* superfamily. *MYB* superfamily proteins are characterized by a highly conserved *MYB* domain at their N-terminus, which is usually composed of 1–4 adjacent imperfect tandem repeats. *MYB* proteins can be divided into different classes depending on the number of *MYB* imperfect tandem repeats (Rs): *4R*-*MYB* proteins (having four Rs), *R1R2R3*-*MYB* proteins (*3R-MYB*, having three Rs), *R2R3*-*MYB* proteins (having two Rs), and *MYB*-related proteins (*1R*-*MYB*, having one R) [5].

The first plant *MYB* gene, C1, was isolated in maize in the 1980s [6]. Since then, different aspects of the *MYB* gene family, such as gene number, sequence characterization, evolution, and functions, have been broadly studied in plants [7,8,9,10,11,12,13,14,15,16]. Among the four classes, *4R*-*MYB* is the smallest. Each gene in this class contains four R1/R2 repeats, but little is known about these proteins in plants [4]. The *3R*-*MYB* proteins have been found in most eukaryotic genomes, and these proteins play a role in controlling the progression of mitosis and cytokinesis [17]. Among these four classes, *R2R3*-*MYB* is the largest in higher plants and is thought to have evolved from the *R1R2R3*-*MYB* gene ancestors by losing the sequences encoding R1 repeats [18]. However, it has also been proposed that *3R*-*MYB* genes are derived from *R2R3*-*MYB* genes by obtaining sequences encoding the R1 repeat through ancient intragenic duplication [19]. Higher plants have numerous *R2R3*-*MYB* genes, which are in accordance with their multiple roles in life processes [20]. The *R2R3-MYB* gene family is widely involved in the regulation of various biological processes, including secondary metabolism [21,22], cell fate and identity [23,24], and organ development [25,26,27]; in addition, most *R2R3-MYB* genes have been suggested to regulate plant responses to biotic and abiotic stress conditions [28,29,30,31,32,33,34]. The fourth class of the *MYB* family contains a single or partial *MYB* repeat and is mainly involved in the control of cellular morphogenesis, secondary metabolism, and circadian rhythm [35,36]. Overall, *MYB*s have diverse functions and play an important role in plant stress tolerance, plant growth, and development.

Extensive studies on various plant species have improved our understanding of the *MYB* gene family in different species [7,8,9,10,11,12,13,14,15,16]. However, a genome-wide analysis of the *MYB* gene family is mainly from studies in green-lineage plants, and little is known about red algae. Currently, only three *R2R3-MYB* genes and five *MYB*-related genes have been identified in *Cyanidioschyzon merolae* [30,37], three *R2R3-MYB* genes in *Chondrus crispus*, three *R2R3-MYB* genes in *Galdieria sulphuraria*, and three *R2R3-MYB* genes in *Porphyra umbilicalis* [30]. Moreover, in these studies, red algae MYB genes have been used as an outgroup of research related to green plants and have never been the protagonist of research. In short, systematic and comprehensive studies on the *MYB* gene family in algae are lacking.

*Pyropia yezoensis* has interesting biological characteristics that have attracted the attention of researchers [38]. For example, *Py. yezoensis* has a complex life cycle alternating between two independent complex multicellular stages that are morphologically distinct: the gametophyte (haploid) and the sporophyte (diploid). The alternating heteromorphic generations might have evolved to respond to different inorganic carbon availability [39]. The edible, leafy thallus (gametophyte) inhabits the upper intertidal zone of exposed coasts, suffering fluctuating environmental stress, such as extreme drought and high temperature [38,39]. For this reason, the *Py. yezoensis* thallus, which is capable of tolerating broad and extreme environmental stress, is of interest as a model for studying the molecular mechanisms of stress resistance. The high-quality chromosomal genome of *Py. yezoensis* and available transcriptomic data also provide convenient resources and lay the foundation for the complete identification of *MYB* genes in *Py. yezoensis* [39,40,41,42].

*MYB*s are the key to regulatory networks controlling development and responses to abiotic and biotic stress, and *Py. yezoensis* has unique physiological and growth characteristics. This study was conducted to identify the *MYB* gene members of *Py. yezoensis* (*PyMYB*s) and to fully investigate their family features and expression profile characteristics. In this study, all non-redundant members of the *MYB* genes in *Py. yezoensis* were screened from available, high-quality, chromosomal-level genomes. We determined the characteristics of *PyMYB* genes based on their physicochemical properties, genomic locations, conserved motifs, and promoters. Then, we analyzed the phylogenetic relationships of these genes combined with *MYB*s in seven other red algae species, three green algae species, and one Glaucophyta species. In addition, the expression levels of the nori *MYB* genes in two life cycle stages and different blade parts, as well as in response to dehydration, temperature, and disease infection stress, were analyzed. Our findings will provide useful resources for future research on the functions of algae *MYB* genes and help us understand the evolutionary history of *MYB* genes in different species.

## 2. Results

### 2.1. Genome-Wide Identification of PyMYB Genes in Py. Yezoensis

A total of 41 candidate sequences containing *MYB* or *MYB*-like domains were identified in *Py. yezoensis*. Twenty-five *PyMYB* candidates were discarded because they lacked a complete MYB domain or complete gene structure in the confirmatory step. The putative novel *MYB* sequences yielded a dataset of 16 nonredundant *MYB* genes containing 3 *R2R3*-*MYB* proteins and 13 *MYB*-related proteins (Table 1). All 16 *MYB* sequences were designated as *PyMYB1–3* (*R2R3*-*MYB*) or *PyMYB4–16* (*MYB*-related genes), according to their classification and genomic distribution in *Py. yezoensis*. Table 1 lists the gene ID, chromosomal location, gene length (ranging from 1394 to 6446 bp), number of introns (ranging from 0 to 2), number of amino acids (ranging from 263 to 1693 amino acids), isoelectric point (pI, ranging from 5.25 to 11.05), molecular weight (MW, ranging from 25.75 to 169.47 kDa), and instability index (ranging from 42.03 to 71.18). Among the 16 *PyMYB* genes, most lacked introns, accounting for half of the *PyMYB* genes, followed by the *PyMYB* genes with one intron and two introns (six and two, respectively) (Table 1, Figure 1A). On the basis of the starting positions of the *MYB* genes within the chromosomes, the 16 *PyMYB* genes were unevenly distributed among the three nori chromosomes (Figure 1B). The number of *PyMYB* genes on chromosomes 1, 2, and 3 was five, seven, and four, respectively; among them, *R2R3*-*MYB* was only present on chromosomes 2 and 3 (Table 1). With the same strategy, a total of 171 *MYB*-related genes, 78 *R2R3-MYB*s, 9 *3R-MYB*s, and 3 *4R-MYB*s, were also identified from the other seven red algae species (*Porphyra umbilicalis*, *Chondrus crispus*, *Cyanidioschyzon merolae*, *Gracilariopsis chorda*, *Porphyridium purpureum*, and *Galdieria sulphuraria*), three green algae species (*Chlamydomonas reinhardtii*, *Micromonas pusilla*, and *Chara braunii*, and one Glaucophyta species (*Cyanophora paradoxa*) (Appendix A).

### 2.2. Phylogenetic Analysis of PyMYB Genes

Because the sequences of R2R3-MYB and MYB-related proteins outside the MYB-binding domains were highly diverse, only the MYB-binding domains (Rs) were used to reconstruct phylogenetic trees.

First, the ML tree was built on the basis of 206 R2R3-MYB protein sequences from 13 representative species of different taxonomic groups, including eight red algae species, three green algae species, one Glaucophyta species, and *Arabidopsis thaliana* (Appendix A). The *R2R3-MYB* genes were then divided into 13 subgroups with robust bootstrap (generally ≥60%): G1–G13 (Appendix A). Overall, the *MYB* genes of eight red algae species were located in most basal clades in the phylogenetic tree, suggesting their ancient origin. Red algae *R2R3-MYB* genes were classified as G2, G3, G4, G5, G7, G9, G10, G11, G12, and G13. Except for genes in G2, G3, and G7, other *R2R3-MYB* genes of red algae were grouped with genes of green plants or glaucophytes. The *R2R3-PyMYB*s belonged to the G4, G5, and G12 subgroups.

An ML tree was also built on the basis of 247 MYB-related protein sequences from 13 representative species from different taxa (Appendix A). These *MYB*-related genes were divided into subgroups with robust bootstraps (generally ≥60%): CCA1-like/R-R-type, TBP-like, I-box-like, TRF-like, and CPC-like subgroups, which have been well-grouped in previous studies [37,43] and seven new groups (NG1–NG7) (Appendix A). The I-box-like and TRF-like subgroups did not contain any red algae *MYB*-related genes, which is consistent with a previous study [37]. In addition, the CCA1-like/R-R-type, TBP-like, CPC-like, NG1, NG2, NG4, NG5, and NG6 subgroups contained red algae *MYB*-related genes and genes from other Archaeplastida lineages, particularly genes from glaucophytes, suggesting their ancient origin; the *MYB*-related *PyMYB*s were scattered in almost all these subgroups except for NG6 and NG7.

Inevitably, through a comparative analysis of two types of ML trees, the topological structure was not very fixed in some genes, including GsuXP005704452, GsuXP_005703373, CcXP_005713040, and CcXP_005717760; in this study, they were finally grouped by subsequent motif composition analysis. In the future, the classification will be improved with the continuous enrichment of red algae genomes and the addition of more red algae *MYB* genes for analysis.

### 2.3. Motif Structure and Synteny Analysis

The similarities in motif organization among genes can be used as evidence to support their common origins. Ten conserved motifs were identified in the MYB proteins using MEME software (https://meme-suite.org/meme/ accessed on 10 October 2023) (Figure 2 and Figure 3, Table 2). Motifs 1 and 4 encoded the MYB DNA-binding domain, and the number of these two motifs in the *MYB* genes of red algae was consistent with their classification in the identification pipeline. In general, *MYB* genes in the same subgroup shared a similar motif composition, and this phenomenon was more obvious in the adjacent genes of the species that were closely related, such as *PyMYB2* and *Ph05827*/*PuOSX69724*, *PyMYB3* and *Ph04980*, *PyMYB4* and *Ph09887*, *PyMYB5* and *Ph02171*/*PuOSX76044*, *PyMYB6* and *Ph02359*, *PyMYB7* and *Ph08492*, *PyMYB8* and *Ph06857*/*PuOSX75147*, *PyMYB9* and *Ph07942*, *PyMYB12* and *Ph05957*, *PyMYB14* and *Ph05025*, and *PyMYB16* and *Ph09161*. The similarity in their motif composition may imply that they originated from the same duplication event and that these genes’ functions might be conservative and important. However, these gene pairs with similar motif compositions had different gene structures, indicating that each species also experienced species-specific evolutionary events.

A synteny analysis was performed between *Py. yezoensis* and other red algae species at the genome-wide level. Most *PyMYB* (13/16) genes had corresponding orthologous genes in other red algae species (Appendix A) and mostly overlapped with the aforementioned gene pairs with similar motif compositions. Thirteen and five *PyMYB* genes were orthologous to the *MYB* genes in *Py. haitanensis* and *Po. umbilicalis*, respectively (Figure 4). No orthologous pairs were found in other red algae species pairs using MCScanX software (https://github.com/wyp1125/MCScanX accessed on 10 October 2023) or gene duplication identification criteria, which may have been caused by the long period of species divergence, leading to larger genetic differences. All identified collinear gene pairs exhibited Ka/Ks ratio values that were far less than one (Appendix A), confirming that the evolution of the *MYB* gene family in the Bangiaceae family underwent a strong purifying selection.

No paralogous duplication events were found in the *PyMYB* genes of the *Py. yezoensis* genome. The same phenomenon also occurred in the other red algae species, except for *C. merolae* and *P. purpureum*, which both contained a pair of *MYB* paralogous genes (CmXP_005537110–CmXP_005536719 and PpKAA8490551–PpKAA8492070) (Figure 3).

### 2.4. Cis-Regulatory Element Analysis of the PyMYB Gene Family

*Cis*-elements can provide important insight into gene function prediction because they regulate gene expression. Therefore, we analyzed the *cis*-elements in the promoter regions of 16 *PyMYB* genes and explored the regulatory mechanisms of *PyMYB* genes in stress response and growth and development (Figure 5). A total of 28 *cis*-elements were identified and mainly grouped into four categories: growth and development, stress response, phytohormone response, and light response (Figure 5). The stress-responsive category accounted for the highest proportion (35.88%) among the four *cis*-element categories, which included ARE (anaerobic induction), DRE (dehydration, low temperature, and salt stress responsiveness), GC-motif (anoxic specific inducibility), LTR (low-temperature responsiveness), MBS (drought inducibility), and TC-rich repeats (defense and stress responsiveness). We found that 31.62% of these elements belonged to the phytohormone response category and were involved in ABA responsiveness (ABRE), auxin responsiveness (TGA element), gibberellin responsiveness (GARE and P-box), MeJA responsiveness (CGTCA motif and TGACG motif), and SA responsiveness (TCA element) (Figure 5). In the light response category, we also found that the ACE motifs, G-box, and Sp1 accounted for 29.31% of all elements (Figure 5). In the growth and development category, we identified four other elements (CAT-box, MAS-like, NON-box, and O2-site), which accounted for 3.20% of all elements (Figure 5). According to these results, *PyMYB* genes participate in diverse functions including phytohormone response, stress response, light response, and plant growth and development.

### 2.5. Expression Patterns of PyMYB Genes in Different Life Stages and Blade Parts of Py. yezoensis

*Py. yezoensis* has a heteromorphic life cycle with two life cycle stages: gametophytic haploid blades (GAM) and sporophytic diploid filaments (conchocelis, SPO). The expression patterns of the 16 *PyMYB* genes in the two different life cycle stages are presented in Figure 6A. *PyMYB15* and *PyMYB16* exhibited the same expression pattern: high expression levels in GAM and low expression levels in SPO. The other *PyMYB* genes exhibited opposite expression patterns.

We also investigated the expression of *PyMYB* expression in the base of blades (BASE), spermatangium cells (SC), carposporangium cells (CC), and vegetative cells (VC) (Figure 6B,C). The primers of 10/16 *PyMYB*s were well designed, and we performed a quantitative real-time polymerase chain reaction (qRT–PCR) analysis; in contrast, the other six genes did not have suitable primers designed for qRT–PCR. These 10 genes were all highly expressed in SC and CC but maintained low or no expression in VC and BASE. However, these *PyMYB* genes exhibited different patterns in the two types of germ cells. *PyMYB3* and *PyMYB5* were relatively highly expressed in CC, whereas *PyMYB4*, *PyMYB7*, *PyMYB8*, *PyMYB9*, and *PyMYB16* were relatively highly expressed in SC.

### 2.6. Expression Patterns of PyMYB Genes in Response to Biotic and Abiotic Stress

We investigated the expression patterns of *MYB* family members in *Py. yezoensis* in response to biotic and abiotic stress to further understand the impact of dehydration and rehydration, high and low temperatures, and red rot disease infection. In dehydration treatments (Figure 7A), *PyMYB3*, *PyMYB7*, and *PyMYB15* exhibited low expression to different degrees during dehydration and high expression during rehydration. The expression of the six *PyMYB* genes, namely *PyMYB1*, *PyMYB6*, *PyMYB10*, *PyMYB11*, *PyMYB13*, and *PyMYB16*, showed no obvious change at the beginning of dehydration stress, whereas their expression levels significantly decreased in severe dehydration treatment (absolute water content 20%, AWC20) and recovered in the rehydration treatment. *PyMYB2*, *PyMYB5*, and *PyMYB9* exhibited higher expression in severe dehydration treatment (AWC20) compared with other treatments. *PyMYB4*, *PyMYB8*, *PyMYB12*, and *PyMYB14* exhibited higher expression in the control treatment (AWC100) compared with the other treatments, indicating that these genes were downregulated under drought stress.

We found different expression patterns in different genes as a result of temperature stress (Figure 7B). Compared with the normal temperature treatment (NT), the expression pattern of the *PyMYB* genes varied in response to high (HT), cold (CS), and freezing (FS) temperature stress. The majority exhibited higher expression under low temperatures (CS treatment or FS treatment) compared with NT and HT treatments, except for *PyMYB1*, *PyMYB3*, *PyMYB6*, *PyMYB7*, and *PyMYB10*. *PyMYB1* exhibited higher expression under HT treatment compared with the other temperature treatments. *PyMYB3* and *PyMYB6* exhibited higher expression under the CS and HT treatments compared with the other two temperature treatments. *PyMYB7* exhibited higher and a more stable expression under CS, NT, and HT treatments compared with FS treatment. *PyMYB10* exhibited higher expression under NT treatment but low expression under other temperature treatments.

The necrotrophic oomycete pathogen *Pythium porphyrae* causes red rot disease, which is lethal to the gametophytes of *Py. yezoensis*. These *PyMYB* genes exhibited significant specific expression under different infection levels (Figure 7C). *PyMYB1*, *PyMYB11*, *PyMYB15*, and *PyMYB16* exhibited high expression only in the control area (CON) but stable low expression in the other two infected areas. *PyMYB2*, *PyMYB7*, and *PyMYB12* exhibited high expression only in slightly infected areas, whereas *PyMYB5* and *PyMYB8* exhibited high expression only in severely infected areas. *PyMYB3*, *PyMYB13*, and *PyMYB14* exhibited high and a more stable expression in control areas and slightly infected areas but low expression in severely infected areas. *PyMYB4*, *PyMYB6*, *PyMYB9*, and *PyMYB10* exhibited low expression in normal areas and high and a more stable expression in the other two infected areas.

## 3. Discussion

### 3.1. Evolution Analysis of MYB Genes in Red Algae

*MYB* TFs constitute one of the largest TF families in the plant kingdom [3]. In this study, we identified 16 *MYB* family genes in the *Py. yezoensis* genome, including 3 *R2R3-MYB*s and 13 *MYB*-related members. We also identified *MYB* genes in seven other red algae species, all of which exhibited significantly lower numbers when compared to higher plants [30,44]. The possible reasons for this difference included the following: red algae lineages underwent genome reduction in their last common ancestor [45,46], and paralogous duplication events were extremely rare in these red algae, although they occurred frequently in higher plants [14,47]. Similar results were also found in some Chlorophyta and Charophyta species with similarly simple organism structures (Appendix A), which may be because higher plants generally require more genes to regulate complicated growth and biological processes [48].

Our analysis of phylogenetic relationships and consensus motifs supports the classification of red algae *R2R3-MYB* genes into 10 subgroups and red algae *MYB*-related genes into 10 subgroups. Most red algae *MYB* genes were clustered with green plants (G4, G5, G9, G10, G11, G12, CCA1-like/R-R-type, TBP-like, and CPC-like subgroups NG1, NG2, and NG4–NG6) or Glaucophyta species (G11, G12, G13, CCA1-like/R-R-type, TBP-like, and CPC-like subgroups NG2 and NG4). These genes were possibly formed in the ancestor of Archaeplastida. There were also some red algae-specific subgroups, such as G2, G3, G7, and NG3, indicating that these types of *MYB* genes may have been lost in other lineages in the very early stages of evolution.

Almost all *MYB* genes in Bangiaceae species, including Py. yezoensis, contain the Gly-rich motif (Motif7), which is absent in Florideophyceae species. This phenomenon was consistent with previous genome research in which it was found that Gly and Ala contained the least carbon atoms (two and three, respectively) and were enriched in Bangiales compared with Florideophyceae [39,49], indicating the evolutionary adaptation to carbon limitation and driving the high GC content evolution. However, a similar GC-rich motif has not been found in other studies of Bangiaceae red algae gene families [50,51]. The *MYB* genes in Bangiaceae species may have evolved directly in response to the effects of carbon limitation. The Gly-rich motif, which controls protein conformation, plays an essential role in the interaction with enzymes, substrates, and other types of non-covalent molecular binding [52,53] and is presumably involved in cold tolerance and acclimation development [54,55].

### 3.2. Functional Diversity of PyMYB Genes

Several biological processes, such as development, stress response, and hormone response, are controlled and regulated by the transcription of gene expression, which involves the molecular switches in *cis*-regulatory elements [56]. Therefore, to explore the functions of *MYB*s, *cis*-regulatory element predictions have been used extensively in several species [9,10]. The results of this study found various development-related, light-responsive, phytohormone-responsive, and stress-responsive elements in the promoter regions of *PyMYB* genes. For example, ABRE, which is involved in the ABA responsiveness of various stressors and plant growth and development [57], was present in all *PyMYB* promoter sequences. The DRE and LTR elements, which were both associated with various abiotic stressors, such as dehydration, low temperature, and salt stress [58,59], were also found in most of the *PyMYB* promoter sequences. As a result, *PyMYB* genes could regulate diverse development and stress-responsive processes.

Spatiotemporal regulation of the *MYB* gene family has been observed in various tissues and developmental stages [37,60,61,62]. Previous studies have shown that some *MYB* genes play an essential role in germ cell specification by activating a germline-specific differentiation program [63]. For example, *Arabidopsis* male germline-specific MYB protein DUO1, as a positive regulator of male germline development, is required for correct male germ cell differentiation, division, and fertilization [64,65]; *MpFGMYB* has been proven to be a key regulator of female sexual differentiation in the haploid-dominant dioicous liverwort, and its loss resulted in female-to-male sex conversion [66]. *Py. yezoensis* is monecious, which means it can produce spermatangium cells and carposporangium cells in one thallus by cell differentiation. In our study, most *PyMYB* genes exhibited significantly high expression in the two types of germ cells compared with vegetative and basal cells, indicating that they played an essential role in cell differentiation. However, the different expression patterns of these genes in the two types of germ cells further suggested that they might play different roles during cell differentiation. *MYB* expression differentiation was found in plant life cycles that alternate between haploid gametophytes (in which male and female gametes are produced) and diploid sporophytes (in which haploid spores are produced through meiosis) [67,68]. The haploid–diploid heteromorphic life cycle of *Py. yezoensis* features a diploid microscopic filamentous sporophyte and a haploid macroscopic blade-forming gametophyte [39]. The results of our study showed the spatiotemporal regulation of the *PyMYB* genes in different life cycle stages of *Py. yezoensis*. Some of the *PyMYB* genes were highly expressed in sporophytes, while others were highly expressed in gametophytes. How these *PyMYB*s performed during development revealed that *Py. yezoensis* regulated the expression of *PyMYB* genes by developing a sophisticated mechanism when and where required.

Major abiotic stressors that can affect the development and growth of laver under various intertidal conditions include dehydration and temperature changes [39]. *MYB* TFs function as key mediators of various stress responses through complex activities, including ABA and other multiple stress signaling pathways [29,69,70] and many other cellular processes [71]. For example, the overexpression of *LlMYB3* in *A. thaliana* L. transgenic plants revealed ABA hypersensitivity and enhanced tolerance to cold, drought, and salt stress [72]. Another experiment on the overexpression of *CmMYB2* yielded similar results [73]. Some *MYB* genes can control secondary wall thickening and participate in stress responses [21,74]. *Py. yezoensis* is an ideal and attractive model for studying the molecular mechanisms of stress resistance because of the complexity of its intertidal living environment and its economic value [39]. In this study, the expression profiles of *PyMYB* genes under biotic (red rot disease infection) and abiotic (dehydration/rehydration and high/low temperature) stress showed that several *MYB* genes were induced significantly and revealed differential expression patterns. These results indicated that different *PyMYB* genes may affect the response to biotic and abiotic stress. For example, *PyMYB2*, *PyMYB5*, and *PyMYB9* responded to dehydration and low temperature by elevating their expression levels, which means that they are closely associated with various abiotic stressors. *PyMYB5* was also upregulated after *Pythium porphyrae* infection, indicating that this gene plays an essential role in the red rot disease response and tolerance and can be used as a candidate gene for future genetic improvement. These results support further functional exploration.

## 4. Materials and Methods

### 4.1. Identification and Characteristics of MYB Genes in Py. yezoensis

To identify the *MYB* gene family members in *Py. yezoensis*, we followed two approaches. First, all of the known *Arabidopsis MYB* gene sequences were used as queries to perform multiple database searches against the protein and genome files downloaded from the *Py. yezoensis* database (https://www.ncbi.nlm.nih.gov/bioproject/PRJNA589917, accessed on 6 September 2022) [39]. The *MYB* genes of *Arabidopsis thaliana* were obtained from online published work and further used in the following phylogenetic analysis [44]. Stand-alone versions of the Basic Local Alignment Search Tool (http://blast.ncbi.nlm.nih.gov, accessed on 6 September 2022) were used, with an e-value cutoff of 1 × 10^−5^. Second, the hidden Markov model profile for the MYB-binding domain (PF00249) was downloaded from the Pfam database (http://pfam.xfam.org/, accessed on 7 September 2022) and used as a query to search the *Py. yezoensis* database using HMMER 3.1 (http://hmmer.org/download.html, accessed on 7 September 2022) with a cutoff value of 0.01. Two more steps were performed to confirm the accuracy of the domains. First, the SMART website (http://smart.embl-heidelberg.de, accessed on 8 September 2022), NCBI-CDD (http://www.ncbi.nlm.nih.gov/Structure/cdd/wrpsb.cgi, accessed on 8 September 2022), and InterProScan (http://www.ebi.ac.uk/interpro/result/InterProScan/, accessed on 8 September 2022) were used to confirm the obtained MYB protein sequences. Then, GARP-like TFs, which contain a consensus sequence (SHLQKY) and are often confused with MYB-related CCA1-like proteins (SHAQK(Y/F)F), should be further removed [37]. GARP-like TFs contain only one of the three regularly spaced Trp residues in the MYB domain. Thus, MYB domains with more than two conserved Trp residues mutated were manually removed to exclude GARP proteins [37,44,47].

All non-redundant *Py. yezoensis MYB* genes were classified into different types (*R2R3*, *MYB*-related) by manually inspecting the number of MYB-binding domains and were named first by their category (*R2R3* and *MYB*-related) and then by genomic location. With the use of the same method described above, *MYB* genes were also identified in seven other red algae species (*Pyropia haitanensis* [42], *Porphyra umbilicalis* [75], *Chondrus crispus* [76], *Cyanidioschyzon merolae* [77], *Gracilariopsis chorda* [78], *Porphyridium purpureum* [79], and *Galdieria sulphuraria* [80]), three green algae species (*Chlamydomonas reinhardtii* (NCBI accession: GCF_000002595.1)), *Micromonas pusilla* (GCF_000151265.2), and *Chara braunii* (GCA_003427395.1), and one Glaucophyta species (*Cyanophora paradoxa* [81]), which are well-studied and evolutionarily important representative species.

We obtained gene structure information of the *PyMYB* genes from the *Py. yezoensis* genome database and displayed it using EVOLVIEW (https://www.evolgenius.info/evolview/, accessed on 10 April 2023). To obtain the protein MW and pI of each *PyMYB* gene, we used ExPASy [82]. The conserved motifs of the PyMYB protein sequences and seven other red algae species’ MYB protein sequences were predicted using MEME (http://meme-suite.org/tools/meme, accessed on 10 April 2023) with the following parameters: an optimum motif width of ≥35 and ≤100 and a maximum number of motifs of 10 [83].

### 4.2. Phylogenetic Analysis of MYB Proteins

We retrieved the amino acid sequences of the MYB-binding domains (R repeats) from each MYB protein sequence and conducted multiple sequence alignments using MUSCLE (v 3.8.425) [84]. The phylogenetic trees of *R2R3-MYB* and *MYB*-related genes were reconstructed with the maximum likelihood (ML) using IQ-TREE multicore version 2.2.2.3 software [85] (parameter: –bb 1000 –TEST). For each reconstruction, we used the –TEST parameter to select the best model and completed 1000 ultrafast bootstraps. We also inferred ML phylogenies for 13 representative species from different taxonomic groups using FastTree (http://www.microbesonline.org/fasttree/, accessed on 17 April 2023).

### 4.3. Gene Synteny and Duplication for MYB Genes

We used MCScanX software (https://github.com/wyp1125/MCScanX, accessed on 10 October 2022) to perform a microsynteny analysis of *Py. yezoensis* and other red algae species [86]. To analyze potential gene duplications, we used the following criteria: (1) the similarity of the aligned gene regions was ≥70% and (2) the sequence alignment covered ≥70% of the longer gene [87]. For each duplicated gene pair, we calculated nonsynonymous (Ka) substitution and synonymous (Ks) substitution using the KaKs_Calculator [88].

### 4.4. Cis-Regulatory Element Analysis of PyMYBs

We extracted sequences in the 2000 base pairs (bp) upstream of the start codon (ATG) in the *MYB* genes from the genome of *Py. yezoensis*. For promoter *cis*-acting regulatory element screening, we analyzed the sequences extracted above using the PlantCARE database (http://bioinformatics.psb.ugent.be/webtools/plantcate/html, accessed on 6 November 2022) [89].

### 4.5. Expression Profile Analysis

The transcriptome data of two life cycle stages in *Py. yezoensis* (gametophyte and sporophyte; NCBI SRA: SRR10527930–SRR10527937), osmotic stress (dehydration/rehydration; NCBI BioProject: PRJNA401507), temperature stress (hot/cold/freezing; NCBI BioProject: PRJNA235353), and red rot disease (*Pythium porphyrae* infection; NCBI BioProject: PRJNA560692) conditions of *Py. yezoensis* were obtained from previously published studies [39,40,41,90]. We used FastQC [91] and Trimmomatic [92] to evaluate the quality and trim the low-quality raw sequencing reads. We performed differential gene and transcript expression analyses of the RNA-sequencing experiments using TopHat and Cufflinks [93]. All fragments, per kilobase of transcript per million fragments mapped (FPKM) values of gene expression, were standardized on the row scale and clustered and visualized with TBtools [94]. Genes with low expression levels (FPKM <2) were filtered.

### 4.6. RNA Isolation and qRT–PCR Analysis

The blades of *Py. yezoensis* used in the experiment were obtained from a farm in Rizhao Lanshan, China. The collected materials were cultured in bubbling natural seawater with Provasoli’s enrichment solution medium with 50 µmol photons m^−2^·s^−1^ at 8 ± 1 °C and a 12:12 light:dark photoperiod for 3 days under laboratory conditions. When the blades entered the mature period, different parts of the blades (e.g., the base of blades, spermatangium cells, carposporangium cells, and vegetative cells) were segmented and selected under an anatomical microscope and confirmed under an optical microscope (Figure 6B). For each treatment, we set three biological replicates and collected and placed the samples in liquid nitrogen for RNA isolation. We used the RNeasy Plant Mini Kit (OMEGA) to extract the total RNA according to the manufacturer’s instructions. We used about 1 μg of total RNA to synthesize the first-strand cDNA using a HiScript^®^ III RT SuperMix for qPCR (+gDNA wiper) Kit (Vazyme Biotech Co., Ltd., Nanjing, China). We performed qRT–PCR following previous work [95]. We used the Light-Cycle^®^ 480 Real-Time PCR System to verify the selected genes according to the following cycling conditions: 95 °C for 90 s, followed by 40 cycles of 95 °C for 5 s, 60 °C for 15 s, and 72 °C for 20 s. The reference genes were cystathionine gamma-synthase 1 (CGS1) and ubiquitin-conjugating enzyme (UBC) genes [96]. Appendix A lists the primer sequences. The 2^−ΔΔCt^ method was used to calculate the relative gene expression values using R 3.6.0 software. The significance of differences was analyzed using the R package “ggsignif” (*p* < 0.05).

## 5. Conclusions

In this study, we performed genome-wide identification and analysis of *MYB* genes encoded in the *Py. yezoensis* genome and identified 3 *R2R3*-*MYB*s and 13 *MYB*-related genes. The phylogenetic analysis suggested *PyMYB*s’ ancient origin and no trace of gene duplication events. Considering the other red algae *MYB* gene information, we inferred that the Bangiaceae-only Gly-rich motif might be linked to evolutionary adaptation. We also conducted an expression analysis of *PyMYB* genes in different developmental stages and stress conditions and found that the *MYB* family had a wide expression profile and played essential roles in growth, dehydration, temperature, and disease stress responses. *PyMYB2*, *PyMYB5*, and *PyMYB9* responded to dehydration and low temperatures through elevated expression levels, indicating that they are closely associated with various abiotic stressors. *PyMYB5* was also upregulated after *Pythium porphyrae* infection, indicating that this gene plays an essential role in the red rot disease response and tolerance and can be used as a candidate gene for future genetic improvement. Our results provide new insights into the potential role of *MYB* TFs in specific developmental transitions and stress responses in *Py. yezoensis*, thus laying the foundation for further functional investigations of *PyMYB* genes.

## Figures and Tables

**Figure 1 plants-12-03613-f001:**
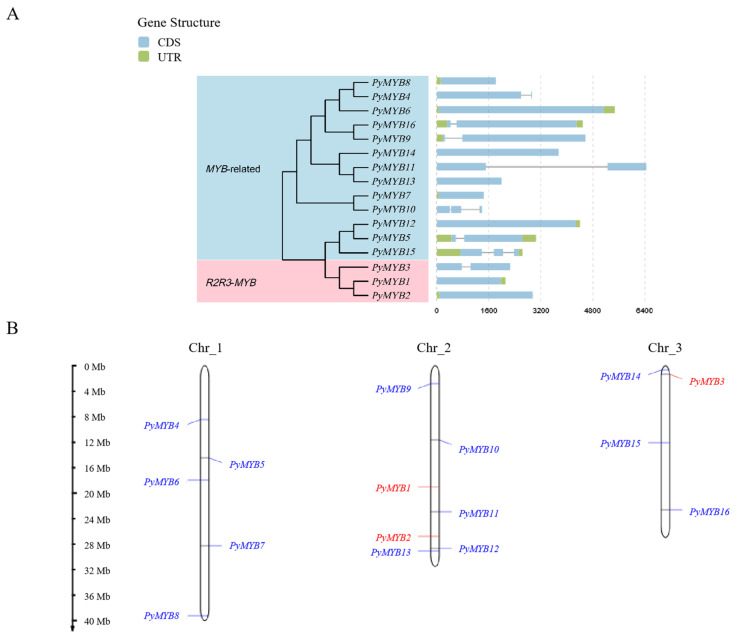
(**A**) Phylogenetic tree and gene structures of *PyMYB* genes. (**B**) Chromosomal location of *PyMYB* genes. Red color, *R2R3-MYB* genes; blue color, *MYB*-related genes.

**Figure 2 plants-12-03613-f002:**
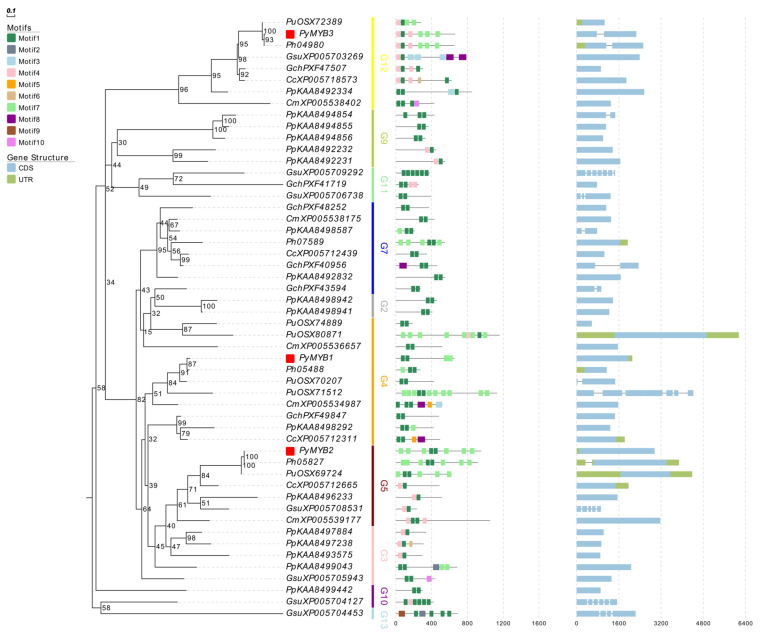
Phylogenetic (**left**) and conserved motif (**middle**) analyses and gene structures (**right**) of eight representative red algae *R2R3-MYB* genes. Red rectangles: *R2R3-PyMYB*s.

**Figure 3 plants-12-03613-f003:**
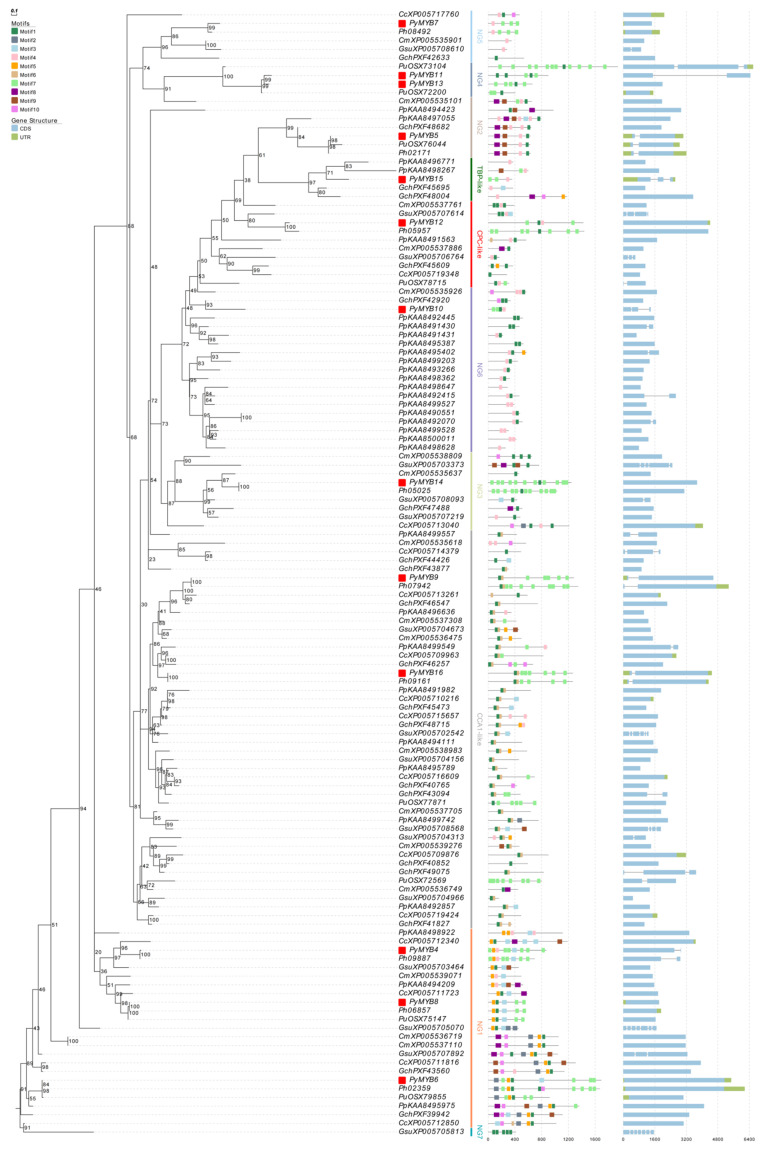
Phylogenetic (**left**) and conserved motif (**middle**) analyses and gene structures (**right**) of eight representative red algae *MYB*-related genes. Red rectangles: *MYB*-related *PyMYB*s.

**Figure 4 plants-12-03613-f004:**
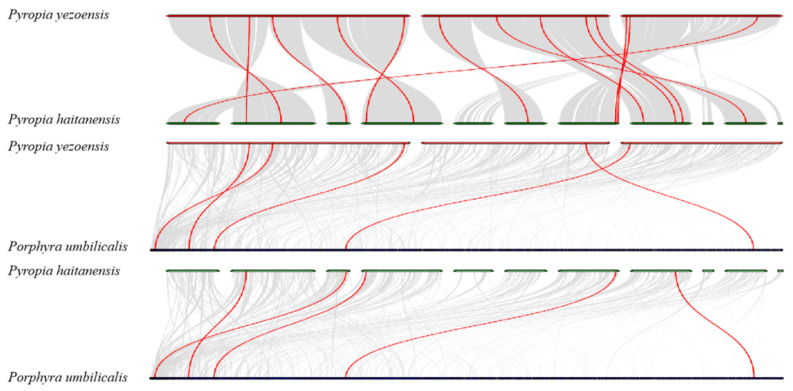
Synteny analysis of *Pyropia yezoensis*, *Pyropia haitanensis*, and *Porphyra umbilicalis*. Gray lines in the background indicate collinear blocks between two genomes, whereas red lines highlight syntenic *MYB* gene pairs.

**Figure 5 plants-12-03613-f005:**
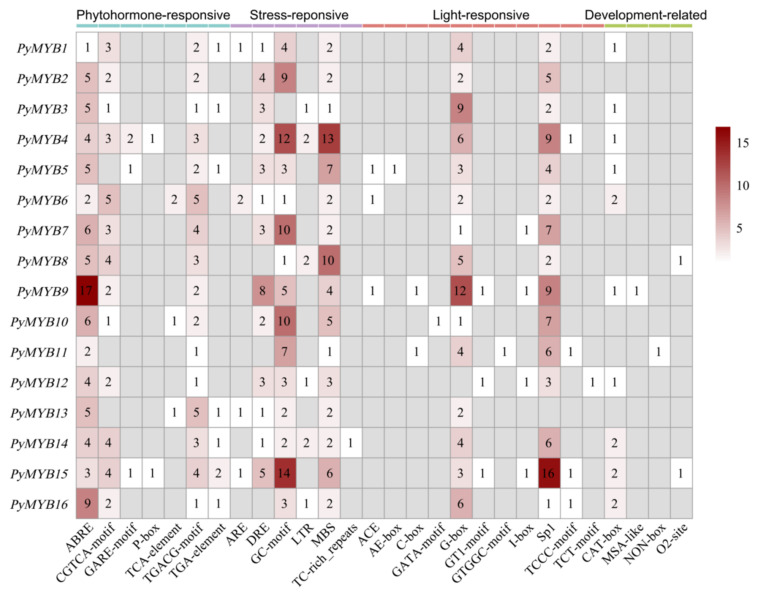
Analysis of 2-kb upstream *cis*-acting elements found in *PyMYB* genes. The different colors and numbers in the grid indicate the number of promoter elements.

**Figure 6 plants-12-03613-f006:**
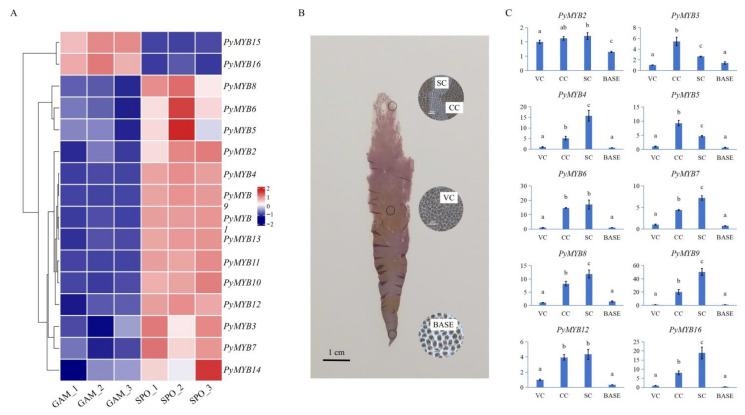
(**A**) Heatmap of the expression pattern of *PyMYB* genes in two life cycle stages: gametophytes (GAM) and sporophytes (SPO). The color bar (right) represents expression data after standardization on the row scale. (**B**) Four parts of *Py. yezoensis* blades. The micrograph in the circle has a scale of 20 μm. (**C**) Expression analysis of *PyMYB*s in different blade parts, as determined by qRT–PCR. The *y*-axis indicates the relative expression level, and the *x*-axis represents the different blade parts collected for expression analysis. The 2^−ΔΔCt^ method was used to calculate the relative gene expression values. Each data point represents the mean ± standard deviation (SD) (*n* = 3). The lowercase letters on the bar chart represented the results of significance analysis (*p* < 0.05).

**Figure 7 plants-12-03613-f007:**
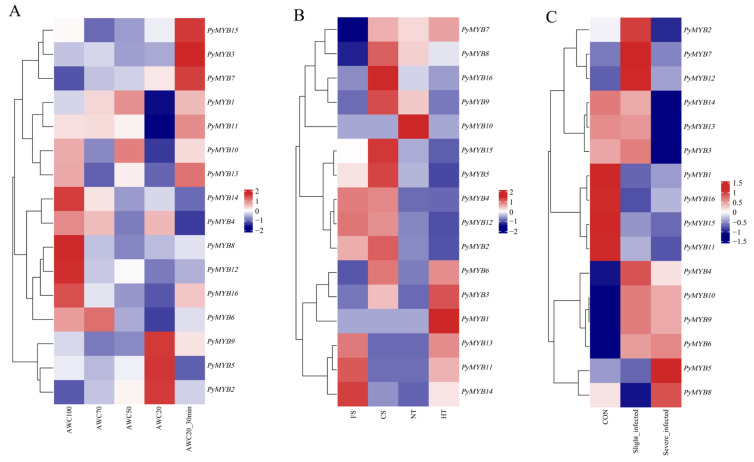
(**A**) Heatmap of the expression patterns of *PyMYB* genes in response to osmotic stress: absolute water content 100% (AWC100, control), absolute water content 70% (AWC70), absolute water content 50% (AWC50), absolute water content 20% (AWC20), and rehydrated 30 min after 20% of water loss (AWC20_30min). The color bar (right) represents log_2_ expression levels (FPKM). (**B**) Heatmap of the expression patterns of *PyMYB* genes in response to temperature stress: normal temperature (NT, 8 °C), high temperature (HT, 24 °C), cold stress (CS, 0 °C), and freezing stress (FS, −8 °C). The color bar (right) represents log_2_ expression levels (FPKM). (**C**) Heatmap of the expression patterns of the *PyMYB* genes in response to red rot disease. “CON” indicates the control (uninfected) area; “Slight_infected” and “Severe_infected” indicate two infected areas with different degrees. The color bar (right) represents expression data after standardization on the row scale.

**Table 1 plants-12-03613-t001:** Characteristics of *PyMYB* genes.

No.	Gene ID	Chr.	Location Coordinates (5′–3′)	Gene Length (bp)	Introns	Protein			Instability Index
						Length (aa)	pI	MW (Da)	
*PyMYB1*	py05834	Chr_2	20,775,567–20,777,675(−)	2109	0	655	9.65	65,595.95	71.18
*PyMYB2*	py09814	Chr_2	29,201,221–29,204,172(−)	2952	0	954	9.71	92,734.27	56.15
*PyMYB3*	py09955	Chr_3	1,412,341–1,414,602(−)	2262	1	663	8.4	65,431.41	46.32
*PyMYB4*	py08181	Chr_1	9,248,952–9,251,564(+)	2613	1	870	6.14	85,547.49	45.30
*PyMYB5*	py00388	Chr_1	15,749,165–15,752,214(−)	3050	1	636	8.17	69,882.18	43.76
*PyMYB6*	py10564	Chr_1	19,596,875–19,602,348(+)	5474	0	1693	5.25	169,473.82	54.09
*PyMYB7*	py03838	Chr_1	30,760,790–30,762,242(−)	1453	0	465	10.37	45,669.49	49.07
*PyMYB8*	py08093	Chr_1	42,710,239–42,712,056(−)	1818	0	562	7.63	59,193.78	44.60
*PyMYB9*	py08650	Chr_2	2,996,682–3,001,254(−)	4573	1	1283	10.17	126,648.01	64.16
*PyMYB10*	py09282	Chr_2	12,729,693–12,731,086(+)	1394	2	263	9.56	25,747.1	53.78
*PyMYB11*	py06564	Chr_2	24,928,598–24,935,043(+)	6446	1	900	8.84	93,136.05	42.03
*PyMYB12*	py07923	Chr_2	31,305,719–31,310,122(+)	4404	0	1425	11.05	143,276.79	55.83
*PyMYB13*	py08311	Chr_2	31,620,311–31,622,302(−)	1992	0	663	10.39	65,237.37	69.35
*PyMYB14*	py05796	Chr_3	726,004–729,747(−)	3744	0	1247	6.67	111,473.37	54.42
*PyMYB15*	py01811	Chr_3	13,184,111–13,186,746(−)	2636	2	360	8.59	33,652.99	68.39
*PyMYB16*	py04398	Chr_3	24,701,888–24,706,378(−)	4491	1	1268	10.28	125,218.85	65.42

**Table 2 plants-12-03613-t002:** Motif information of MYB proteins in eight red algae.

Motifs	Width (aa)	Pfam (*p* < 1 × 10^−5^)	Sequences
Motif1	43	Myb DNA-binding domain	SIKRGWWTEEEDELLLELLKLGGRNWKRIAKHLPGRTDNQCRS
Motif2	69	SWIRM domain	LPEFFLGRFASKTPEVYKQYRBFMIDTWRQDPTRYLTATAVRRHLAGDACAILRVHAFLEHWGLINYGV
Motif3	68	-	GWMPKRGDFDYEWDDEAEEIIADMEILEDDTPEEVELKLRLLEIYNAKLDERERRKEFVLSRNLLDFK
Motif4	44	Myb DNA-binding domain	RRRWTPEEDELLREAVAKYGAGDWARLAAEYLPGRTGKQLRARW
Motif5	47	Zinc finger, ZZ type	IEYCCDCCGDDCSRLRYHCATCADMDLCPDCFSVGRYPSPHKARDFI
Motif6	35	-	HAQKYFLKVQKNKTGEYVPPPRPKRRASSPYPRAA
Motif7	49	-	AGVAGAGAGGGGAAGGGGGAAGAGGGGGGGGGGGGGGGGGGGGGGGGGA
Motif8	82	DnaJ	KLTARNSDDLYELLELGDKRWHATEDDIKKAFRRLSLKYHPDKIAHAGEEAMEDAGEHFKAMRKAFDTLSDRRKRAAYDSID
Motif9	69	-	LGDENTPMEEVHAFYDFWYRFKSWRDFSADLEFDPEQADSREERRWMDRQNAKHIKKRRAAEASRIRLL
Motif10	53	SWIRM-associated domain at the N-terminal	ATHIVYPDPPGTTEAETEGEDYCRALKRKGBQVLVHWWYYPDSYDSWIPRQEV

## Data Availability

All datasets generated in this study are included as Appendix A of this article.

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
