# Peer review of "Genome-Wide Identification and Analysis of MYB Transcription Factors in Pyropia yezoensis"

_plants, 2023, doi:10.3390/plants12203613_

Round 1
Reviewer 1 Report
The authors analysed MYB genes in Py. yezoensis genome-wide and found 3 R2R3-MYBs and 13 MYB-related genes. PyMYBs' ancient origin and lack of gene duplication were indicated by phylogenetic research. The Bangiaceae-only Gly-rich motif may be associated with evolutionary adaptation based on other red algae MYB gene data. They examined PyMYB gene expression in various developmental stages and stress settings and discovered that the MYB family is active in growth, dehydration, temperature, and disease stress responses. The MYB TFs may play a role in key developmental transitions and stress responses in Py. yezoensis, laying the groundwork for future PyMYB gene functional studies.
I proposed that the manuscript be published because it provided novel information about the families of MYB TFs.
Reviewer 2 Report
Dear Esteemed Authors,
Your manuscript delves into an extensive exploration of MYB genes in red algae, a subject of great interest. However, it is crucial to clarify the purpose behind studying this particular species of algae within the manuscript's introduction. Providing a clear rationale for this study will enhance the comprehensibility and relevance of your research. I have provided detailed comments in the attached PDF file

Its easy to follow.
Reviewer 3 Report
This article identified and analyzed 13 A genes and 3 B genes in plants through a series of bioinformatics methods, and found that upregulation of gene 5 expression can to some extent enhance the plant's resistance to abiotic or biological stresses such as drought, low temperature, and bacterial infection.But the article has the following problems.
1. This article conducted traditional bioinformatics analysis to identify genes related to the MYB gene family, but the innovation points are not yet clear, and the improvements made based on previous research have not been elaborated in detail. Therefore, it is recommended to delve into the research innovation points and improve the significance of the research.
2. There are few experiments designed in this article, and most of them are traditional bioinformatics analysis. Bioinformatics analysis is only for prediction and cannot explain the experimental results, resulting in insufficient experimental argumentation. The reliability of the experimental results is questionable. It is recommended to expand our thinking and supplement experiments.
3. The writing of this article is not standardized enough. The genes should be italicized and the protein should be in bold. Please check and correct the entire article, and ensure that the writing is standardized in the images. All genes in the images are not italicized, please correct.
4. In Figure 6, the scale of Figure 6B is not marked, and the uploaded image is blurry. Please upload a clearer image and mark the scale size.
5. The gene expression level analysis in Figure 6C was not significantly analyzed, and some experimental results showed obvious error bars. Whether it is considered that the experimental technical error is significant, resulting in inaccurate experimental results.
6. All pictures uploaded in this article are of insufficient clarity. Please upload a clearer picture.
The English quality of this article is acceptable and meets basic standards.
